# Memorization bias impacts modeling of alternative conformational states of solute carrier membrane proteins with methods from deep learning

G.V.T. Swapna[1,2], Namita Dube[1], Monica J. Roth[2]*, Gaetano T. Montelione [1]*

1 Department of Chemistry and Chemical Biology, Center for Biotechnology and Interdisciplinary Sciences, Rensselaer Polytechnic Institute, Troy, New York, United States of America, 2 Department of Pharmacology, Robert Wood Johnson Medical School, Rutgers, The State University of New Jersey, Piscataway New Jersey, United States of America

* roth@rwjms.rutgers.edu (MJR); monteg3@rpi.edu (GTM)

## Abstract

The Solute Carrier (SLC) superfamily of integral membrane proteins transport a wide array of small molecules across plasma and organelle membranes, and function as important drug transporters and as viral receptors. They populate different conformational states during the solute transport process, including outward-open, intermediate (occluded), and inward-open conformational states. For some SLC proteins this structural "flipping" corresponds to swapping between conformations of their N-terminal and C-terminal symmetry-related sub-structures. Conventional Alpha-Fold2, AlphaFold3, or Evolutionary Scale Modeling methods typically generate models for only one of these multiple conformational states of SLC proteins. While several AI-based protocols for modeling multiple conformational states of proteins have been described recently, these methods are often impacted by "memorization" of one of the alternative conformational states, and do not always provide both the inward- and outward-open conformations of SLC proteins. Here we assess the impact of memorization in modeling SLC proteins with AlphaFold2/3, and describe a combined ESM – template-based-modeling process, based on a previously described template-based modeling method that relies on the internal pseudo-symmetry of many SLC proteins, to consistently model the alternate conformational states of SLC proteins. We also demonstrate how the resulting multi-state models can be validated by comparison with sequence-based evolutionary co-variance data (ECs) that encode information about contacts present in the various conformational states adopted by the protein. This simple, rapid, and robust approach for modeling conformational landscapes of pseudo-symmetric SLC proteins is demonstrated for several integral membrane protein transporters, including SLC35F2 the receptor of a feline leukemia virus envelope protein required for viral entry into eukaryotic cells.

**Data availability statement:** All scripts and key data generated in this study are available at https://doi.org/10.5281/zenodo.17386602

**Funding:** FINANCIAL DISCLOSURE This work was supported financially by National Institutes of Health NIGMS (https://www.nigms.nih.gov/) grants R35 GM141818 (to G.T.M.) and R35 GM122518 (to M.J.R.), and by the Rensselaer Polytechnic Institute (RPI) Bio-computing and Bio-informatics Constellation Chair Fund (to G.T.M). GTM also acknowledges access to the RPI Center for Computational Innovations (CCI) computing infrastructure. Research costs and partial salaries of G.V.T.S. were supported by NIH grants R35 GM141818 and R35 GM122518. Research costs and partial salaries of N.D. were supported by NIH grant R35 GM141818. Research costs and partial salaries of M.J.R. were support by NIH grant R35 GM122518. The funders had no role in study design, data collection and analysis, decision to publish, or preparation of the manuscript.

**Competing interests:** I have read the journal's policy and the authors of this manuscript have the following competing interests: GTM is a founder of Nexomics Biosciences, Inc.

**Abbreviations:** AF2, AlphaFold2 Multimer; AF3, AlphaFold3, EC, evolutionary covariance; ESM, Evolutionary-Scale Modeling, LDDT, local-distance difference test; MD, molecular dynamics; ML, machine learning; mmCIF, macromolecular Crystallographic Information File; MSA, multiple sequence alignment; PDB, Protein Data Bank; pLDDT, predicted Local-Distance Difference Test, a confidence score predicted from ML; TM, Template Modeling score to assess similarity between two protein structures.

## Author summary

The Solute Carrier (SLC) superfamily of integral membrane proteins transport a wide array of small molecules across plasma and organelle membranes, and function as important drug transporters and as viral receptors. We describe a combined ESM – template-based-modeling process that relies on the internal pseudo-symmetry of many SLC proteins, to consistently model the alternate conformational states of SLC proteins. We also demonstrate how the resulting multi-state models can be validated by comparison with sequence-based evolutionary co-variance data (ECs). Using either Alpha-Fold2 or AlphaFold3 in this process reveals bias due to "memorization" that challenges the view that modeling of the multiple conformational states of this important class of integral membrane proteins is a largely solved problem. This simple, rapid, and robust approach for modeling conformational landscapes of pseudo-symmetric SLC proteins is demonstrated for several integral membrane protein transporters.

## Introduction

Proteins adopt multiple conformational states which are essential to their functions. While AlphaFold2/3 (AF2/3) [1], Evolutionary Scale Modeling (ESM) [2], and related machine-learning methods [3,4] can provide accurate structural models of proteins, for systems that adopt multiple conformational states, conventional AF2/3 and ESM calculations often deliver only one of the multiple states observed experimentally [5–13]. Recent advances have been reported using modified AF2 protocols and "enhanced sampling" methods to model multiple conformational states of proteins, including integral membrane proteins [14]. Promising approaches use a conventional AF2 platform with curated input such as (i) state-annotated conformational templates [15,16], (ii) shallow multiple sequence alignments (MSAs) chosen either randomly (AlphaFold-alt) [6,17] or by clustering homologous protein sequences (AF-cluster) [8], (iii) very shallow MSAs and even single protein sequences [9,18] that allow knowledge inherent to the AI to dominate the modeling process, or (iv) using MSAs masked at multiple positions, as implemented in both (SPEACH-AF) [19] and AF-sample2 [12], to bias the prediction toward alternative conformational states. AF2 calculations using network dropouts (AF-sample) can also generate conformational diversity [20–24]. Despite these advances, challenges remain in reliably modeling alternative conformational states that are observed experimentally. In particular, assessments report various enhanced sampling methods to be successful in modeling multiple conformational states for 50% (or less) of experimentally-available alternative conformer pairs [9,11,13]. This observation suggests that at least some cases of successful modeling result from some kind of memorization by the AI, rather than its inherent "learning" of protein structure principles [9,13]. Such memorization can bias the AI-based modeling towards conformational state(s) used in the training

process and prevent accurate modeling of alternative conformational states. To the degree that memorization biases the successful prediction of alternative conformational states, more robust methods leveraging the tools of AI-based modeling are required.

The Solute Carrier (SLC) superfamily of integral membrane proteins function to transport a wide array of solutes across the plasma and organelle membranes. The superfamily includes more than 66 SLC protein families (https://www.biopara-digms.org/slc/intro.htm), each including many individual proteins. SLC proteins transport a wide array of molecules, including sugars, amino acids, vitamins, nucleotides, metals, inorganic ions, organic anions, oligopeptides, and drugs [25–28]. Some are orphan transporters with no known substrate. SLC proteins can also function as receptors for viral entry into the cell [29]. They constitute a major portion of all human transporter-related proteins and play key roles in human health and disease [27,28,30].

Despite being classified as a single superfamily, the various SLC fold families do not share a single common fold classification and are not all phylogenetically related. For example, the two most common SLC fold families, the major facilitator superfamily (MFS) fold, which constitute the largest class of SLC proteins, and the LeuT fold, another important class of SLCs, are topologically and structurally distinct [27]. However, despite these differences, many SLC transporters have a characteristic structural architecture with pseudo two-fold symmetry, where the two halves of the protein structure are related by a two-fold symmetry axis in the plane of the membrane bilayer [27,31]. These halves have a similar fold but non-identical conformations, enabling the protein to adopt multiple conformational states essential for its function. MFS-fold SLC proteins have a "6 + 6" topology comprised of two "inverted pseudo-repeat" 6-helical bundles with antiparallel orientations related by a pseudosymmetry axis, while the strikingly similar but topologically distinct LeuT-fold membrane proteins feature two 5-helical bundles with "inverted pseudo-repeat" sequences that form structures related to one another by a pseudosymmetry axis [27]. Some (but not all) other SLC proteins also have folds with internal structural pseudosymmetry [27].

SLC proteins populate different conformational states during the transport process, including "outward-open", with a surface cavity directed one way, intermediate states (i.e., occluded, with no surface cavity), and "inward-open" with a surface cavity directed to the opposite side of the membrane [26,27]. These "inward-open" and "outward-open" conformational states are sometimes called "inward-facing" and "outward-facing" states in the literature. Crystal structures have been solved for inward-open, occluded, and outward-open states of several MFS and LeuT SLC proteins; for a few SLC proteins both inward and outward-open states have been determined by X-ray crystallography or cryoEM [32–36]. This conformational "flipping" confers an "airlock" or "revolving door" function, which underlies their mechanisms of symporter or antiporter solute transport [26,27,31]. The switch between outward- and inward-open states results from swapping of the conformations of the N-terminal and C-terminal symmetry-related sub-structures, in which the N-terminal helical bundle switches to adopt the conformation of the C-terminal helical bundle, while simultaneously the C-terminal helical bundle switches into the original conformation of the N-terminal helical bundle. These dynamic structural and biophysical properties confer to SLC proteins their functions as gates for symporter and antiporter transport of biochemically-important solutes and biomolecules [27,28].

Computational methods significantly impact our understanding of SLC protein structure-function relationships and can guide experimental design. However, as they are medium-sized integral membrane proteins, molecular dynamics simulations are quite challenging, requiring powerful computing resources, accurate potential energy functions, and appropriate simulation of membrane-mimicking environments. The evolving AI-based enhanced sampling methods outlined above can sometimes provide models of multiple conformational states of SLC proteins, but they are not always successful [6–12]. These observations suggest the need for more robust methods for modeling the multiple conformational states of this important class of membrane protein transporters.

Importantly, multiple conformational state modeling of proteins can be guided by evolutionary covariance (EC) analysis of functionally-preserved direct contacts, which can provide information about contacts present in the two (or more) states

adopted by the protein structure [37–43]. Of special significance for SLC proteins is their unique pseudo-symmetrical transport mechanisms, which provides the basis for an elegant method of modeling the inward-open (or outward-open) conformations of some SLC proteins from knowledge of their outward (or inward) open conformations by swapping the pseudo-symmetric structures of the N- and C-terminal halves, and then using the resulting virtual structure as a template to model the alternative conformational state [31,44–50]. Although it is logical to combine the two concepts of validation with EC-based contact information and swapping of pseudo-symmetric structures, it has not yet been implemented as a general strategy for modeling SLC proteins.

Here we describe a simple and robust approach for modeling alternative conformational states of pseudo-symmetric SLC proteins using a combined ESM – template-based-modeling process inspired by the methods of Forrest and others [31,44–50]. In this approach, templates for alternative conformational states are generated from a reordered, or "flipped virtual sequence", using ESMFold [2], and template-based modeling is then performed using either AF2/3 [1,51] or, where training bias impacts the AF2 structure prediction, with the template-based modeling software MODELLER [52]. First, an ESM-AF approach was used to model the inward-/ outward-open forms of two SLC proteins, human ZnT8 (SLC30A8, a Zn transporter) and *Escherichia coli* D-galactonate:proton symporter (SLC17, a MFS superfamily transporter) for which experimental structures of both outward- and inward-open states are available, and the resulting models of alternative conformations were validated against EC-based contact maps and by comparison against atomic coordinates determined by cryoEM or X-ray crystallography. For two additional SLC proteins, *Zea mays* CMP-sialic acid transporter 1 (SLC35A1) and *Saccharomyces cerevisiae* GDP-mannose sugar transporter 1 (SLC35D subfamily), only the outward-open forms are available as experimental structures. In these cases, AF modeling was found to be biased towards these states and the alternative inward-open forms were modeled with an ESM-MODELLER process. Models were then validated by comparison against EC-based contact maps. For SLC35F2, although neither inward nor outward-open experimental structures are available, the outward-open state is strongly preferred when using conventional AF2. However, the inward-open conformational state could be modeled using either the ESM– AF2 or ESM – MODELLER processes. Both the inward- and outward-open structures were validated against EC-based contact maps. For other SLC proteins where experimental structures are available for only one conformational state, significant bias towards one state or the other was also observed using AF2 or AF3. In these cases, the ESM-MODELLER approach was successful in modeling both inward and outward-open states, which were validated by comparisons against EC-based contact maps.

## Methods

### Evolutionary covariance (EC) - based contact predictions

EC-based contact predictions were performed using evolutionary covariance analysis with *NeBcon* (Neural-network and Bayes-classifier based contact prediction) https://seq2fun.dcmb.med.umich.edu/NeBcon/, a hierarchical algorithm for sequence-based protein contact map prediction [53], with a probability threshold of 0.7. A second server, *EVcouplings server* [37] https://evcouplings.org/ was also used to confirm these contact predictions.

Contact maps for experimental and predicted structures were obtained from *CMview* [54], an interactive contact map visualization and analysis tool. Contact maps were generated for interresidue C$\alpha$ distances of < 10.0 Å. The contact lists generated from protein structure models were then imported into excel spreadsheets for overlay and comparison with the EC-based predicted contacts.

### AlphaFold2, AlphaFold3, ESMfold, and MODELLER modeling

AlphaFold2 (AF2) [1] modeling was performed using ColabFold v1.5.5 server [55] with *AlphaFold2.ipynb* scripts. The standard AF2 modeling in this study used no templates (unless specified), default multiple sequence alignments (MSAs),

recycle of 12, and random dropouts, although other protocols were also assessed. The Amber-relaxed top-ranked model was taken as the final predicted structure. AlphaFold3 (AF3) modeling was performed using the Google Deep Mind/Isomorphic Lab server (https://alphafoldserver.com/welcome) [51] (with no structural templates) or with a locally-installed instance of AF3 (with structural templates), as described for specific cases. Evolutionary Scale Modeling (ESMfold) [2] models were generated using the *ESMFold_advanced.ipynb* colab script. Models were generated with random masking of input sequences (masking_rate = 0.15), stochastic_mode = "LM" no dropout), and recycle of 12. The model with the highest pTM score was selected as the final model. A locally installed version of MODELLER 10.4 [52,56] was used for conventional template-based modeling. For each run, 20 models were generated, and the one with the lowest DOPE (Discrete Optimized Protein Energy score) was selected as the representative structure.

### AlphaFold-alt

Enhanced sampling using shallow MSAs with AlphaFold-alt (AF-alt) was carried out as described by Meiler and co-workers [6], using scripts kindly provided by Dr. Davide Sala and executed on a local cluster with four A100 Nvidia HGX GPUs. In each AF-alt run, 480 models were generated using randomly-sampled shallow MSAs (typically ranging from 16 - 32 sequences, though in some cases described in Supplementary Material runs were made using 4 - 8, 8 - 16, and even with a single sequence), with 30 models created per MSA depth. Each run took < 3 hrs. For each model, disordered N- and C-terminal regions were removed, and the average pLDDT (<pLDDT>) was computed for the remaining residues.

### AF_Sample and AF_Sample2

Massive sampling was carried out using *AF_Sample* and *AF_Sample2* [12,20,21] with AF2, executed on a local cluster of four A100 Nvidia HGX GPU processors, following protocols described elsewhere [24]. *AF_Sample* inferences used *AF-Multimer* model weights v2.1.2, v2.2.0, and v2.3.2, with no templates. Runs with *v2.1.2* used 21 max_recycles, *v2.2.0* used the default of 3, and *v2.3.2* used 9. *AF_Sample2* inferences used the same model weight variations but were run with 3 max_recycles and no templates.

Hydrogen atoms were added to files generated by *AF_Sample, AF_Sample2,* and *AF_Alt* using a custom script which employs the Amber force field, analogous to the method employed by the original AF2 manuscript [1]. These scripts are provided at https://doi.org/10.5281/zenodo.17386602. Each of these enhanced sampling methods can be quite aggressive in generating conformational diversity and also models that are not physically reasonable: e.g., incorrect amino acid chirality, non-native *cis* peptide bonds, and other biophysically incorrect features, particularly in the not-well-packed residue segments of the modeled proteins. The most egregious of these physically unreasonable models were identified and removed. The resulting relaxed models were used for further analysis.

### Statistical methods

Backbone root-mean-squared deviation (RMSD) and global distance test (GDT) scores for structural comparisons were performed using the methods of Zemla implemented on their public server http://linum.proteinmodel.org/ [57].

### Results

The challenge we encountered arises from the fact that conventional AF modeling protocols generally provided only one of the multiple conformations of SLC proteins, particularly when only one of these states was available as an experimental structure at the time of training. Even enhanced sampling methods successfully generate alternative conformational states for only some multistate proteins [6,8–13]. These observations motivate the need for robust and consistent methods for modeling alternative conformational states (outward-open vs inward-open) of SLC proteins, at the very least for use as reference states for assessing the evolving deep learning methods for generating alternative conformational states of proteins.

## Bias of AF2 in modeling alternative conformational states

AF2 bias in modeling alternative conformational states is documented for several SLC proteins used in this study in S1 Table. In the first two systems (DgoT and ZnT8), both inward- and outward-open states were available in the PDB at the time of AF2 training. AF2 with full MSAs (with or without dropouts) is biased towards predicting only the inward-open state. AF2 inference with a single sequence consistently fails to generate a reasonable model. However, using shallow MSAs (4 - 16 sequences) alternative states similar to the experimentally-determined alternative state conformations are delivered, but only as a small fraction of the generated models. For the next 5 systems listed in S1 Table, only one state (inward- or outward-open) was available in the PDB at the time of AF2 training. For this set, AF2 with full MSAs (with or without dropouts) is biased towards predicting only the state available for training (in 4 cases the outward-open state, in 1 case the inward-open state). Again, AF2 inference with only a single sequence consistently fails. Using shallow MSAs the alternative state is delivered as one or more of the generated models in only one of the 5 cases; i.e., for SLC19A1. For the last 3 cases summarized in S1 Table, no homologous structures were available in the PDB at the time of training. AF2 with full MSAs (with or without dropouts) again delivers a single dominant state. AF2 inference with only a single sequence consistently fails. Using shallow MSAs (4 - 16 sequences) the alternative state is delivered as at least one of the generated models for SLC19A2, but only the outward-open state is generated for SLC35F2 or SLC35F3. These results document the challenges AF2 faces in modeling alternative conformational states when one or more state is potentially available in the training data. Overall it was much harder to generate alternative conformational states (i.e., both inward- and outward-open states) for these SLC proteins using shallow MSAs than we expected from published studies.

## ESM-AF/MODELLER protocol

To address these challenges, we reasoned that it might be possible to generate structural templates using alternative deep learning methods, and use these templates to guide the modeling process along the lines that have been demonstrated so successfully by Forrest et al [31,44–50]. The ESM-AF/ ESM-MODELLER process for modeling alternative conformational states of SLC transporters that have structural pseudo-symmetry is outlined in **Fig 1**. It is based conceptually on methods used for other pseudo-symmetric SLC proteins [31,50], in which the pseudo-symmetric halves of the transporter are first identified as an N-terminal protein sequence (blue in **Fig 1**) and C-terminal protein sequence (purple in **Fig 1**), and the N-terminal protein sequence is then modeled using the C-terminal segment as a structural template, and the C-terminal protein sequence is modeled using the N-terminal segment as a structural model. However, application of this method using conventional modeling methods can be challenging if the sequence similarity in these two halves of the protein sequence is low, making it difficult to determine the correct alignment for template-based modeling. In the ESM-AF/MODELLER process, the N-terminal (blue) and C-terminal (purple) segments of protein sequences are first swapped to create a *virtual flipped sequence*. Note that this flipped sequence has no homologs with which to generate a multiple-sequence alignment. The 3D structure of this virtual sequence is then modeled using *ESMfold*, a large-language model-based method that requires no templates and only a single input sequence. The resulting *virtual structure* serves as a structural template for modeling the original protein sequence using template-based modeling with *AF2/3* (if no state-specific bias is observed) or with MODELLER.

In this protocol, *ESMfold* is used to model a *virtual template structure*. S2 Table summarizes tests carried out to compare the effectiveness of *ESMFold* and *AF2* in generating these *virtual template structures*. In all cases, when using single sequences as input *ESMfold* provided a structural template with backbone structure matching the expected alternative conformational state, while AF2 was not able to generate reasonable structures for any of the *virtual flipped sequence*. AF2 could sometimes successfully generate a virtual structure with the alternative conformation when using shallow MSAs as input; however, this is not useful for our protocol which uses a single *virtual flipped sequence*. In our experience *AF2* is less consistent and robust than *ESMfold* in generating a good quality structural template from the *virtual flipped sequence*.

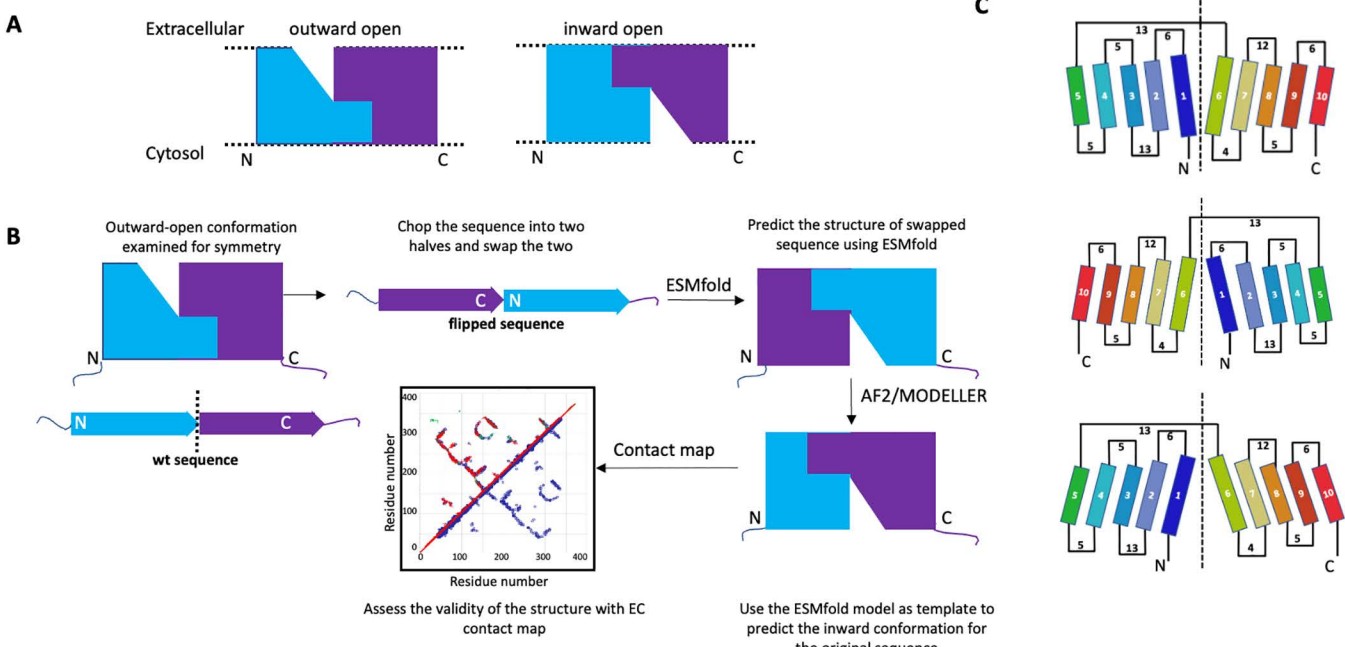

**Fig 1. The ESM-AF/MODELLER protocol for modeling alternative conformational states of pseudo-symmetric SLC proteins.** (A) Cartoon representation of inward/outward-open conformers representing the pseudo-symmetry of the helices, with pseudo-symmetry halves indicated in blue and purple. (B) Protocol to model inward/outward-open conformers for symmetric helical transmembrane proteins (C) Topology diagrams showing the conformational flip of a representative 10-helical SLC protein (SLC35F2). The vertical dotted line represents the symmetry axis of the pseudo-symmetric halves of the SLC protein. Numbers represent the number of residues in the membrane-external loops. The top image represents the outward-open state, the middle image is the *ESMfold* virtual protein structure generated from a *virtual flipped protein sequence*, and the bottom image the inward-open state generated by comparative modeling using the virtual protein structure as a modeling template.

## Validating the ESM-AF2 modeling protocol

As an initial test case of the ESM-AF2 method for modeling alternative conformational states of SLC proteins, we selected human ZnT8 (SLC30A8), a 2 x 320-residue homodimeric integral membrane Zn-transporter, for which structures have been determined by cryoEM [33] (PDB ids: 6xpd, 6xpde, and 6xpf, at resolutions of 3.9 Å, 4.1 Å, and 5.1 Å, respectively). ZnT8 (PDB id: 6xpf) has two subunits; in the absence of Zn, chain-A is in an inward-open conformation and chain-B in an outward-open conformation. The contact maps for the inward- and outward-open states demonstrate that key differences involve interactions between helices H1, H2 and H3 with helices H5 and H6 (Fig 2). Conventional AF2-colab calculations using the standard protocol outlined in the Methods section provided a structure with the inward-open conformation, matching the cryoEM inward-open structure 6xpf-A (Cα RMSD = 2.00 Å) (Fig 2A). We then used the ESM-AF2 modeling protocol outlined in Fig 1 to generate the outward-open conformation and compared it with the experimentally determined outward-open cryoEM structure. The computed outward-open ZnT8 model showed excellent agreement with the experimental 6xpf-B (Cα RMSD = 1.09 Å) (Fig 2B). We compared residue-residue contact maps for the experimental and ESM-AF2 outward-open models against each other and with an EC-based contact map derived from multiple-sequence alignments of ZnT8 homologs (Fig 2C and 2D). The AF2-modeled inward-open structure, has a contact map that is nearly identical to that of the experimental inward-open structure (Fig 2C); the outward-open structure computed using the ESM-AF2 protocol is also essentially identical to the experimental outward-open structure (Fig 2D). While many ECs are common to both the outward- and inward-open conformations, the ECs contain information about both states, and several are unique to each conformation, aligning precisely with the corresponding contacts in the computed models (circled in Fig 2C and 2D). Hence, the ESM-AF2 protocol successfully modeled both

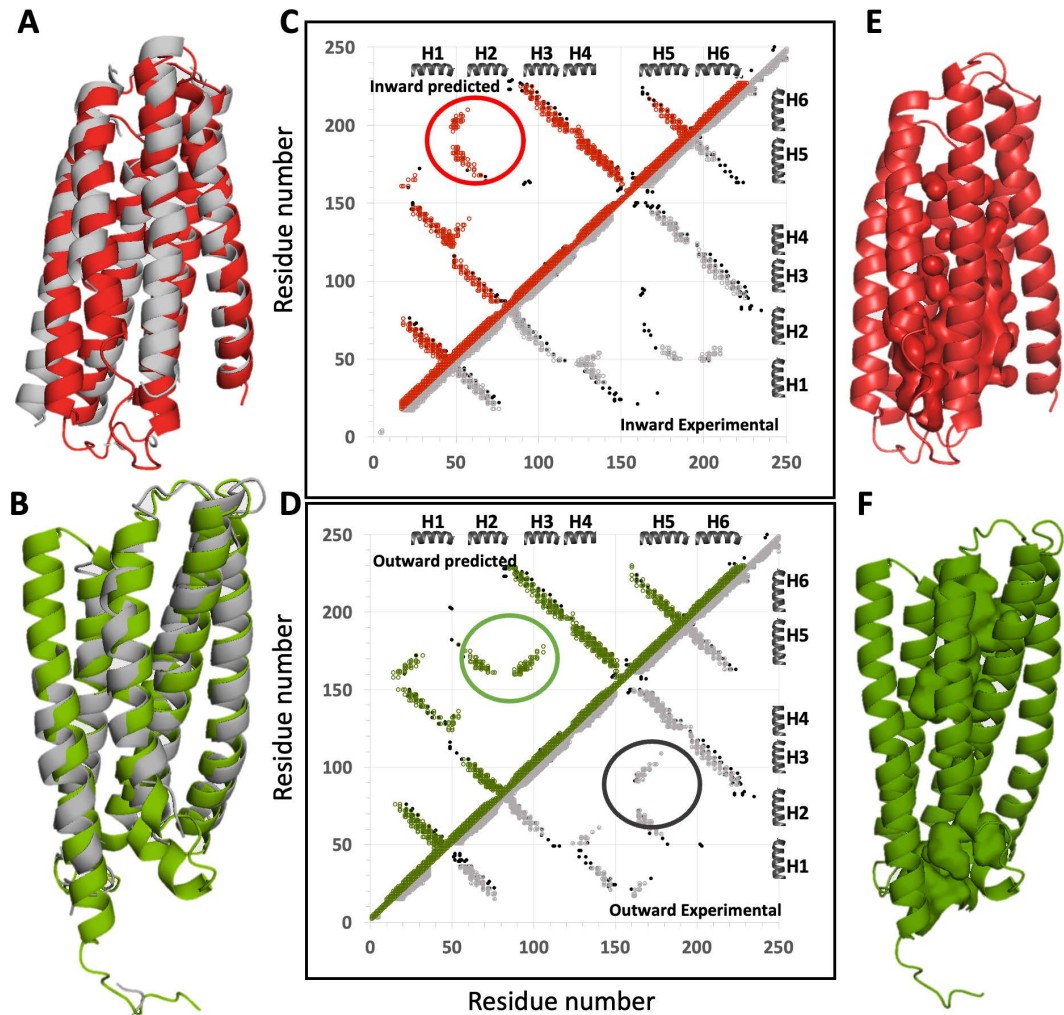

**Fig 2. Validation of ESM-AF2 protocol using an SLC protein with both outward- and inward-open experimental structures.** The cryo-EM structure of human ZnT8 WT in the absence of zinc has two chains, with one subunit in an inward-open conformation and the other in an outward-open conformation (PDB id: 6xpf chain A and B respectively). (A) Superposition of the AF2-predicted (red) and experimental (grey) inward-open structures (C$\alpha$ RMSD = 2.00 Å. (B) Superposition of the ESM-AF2 outward-open model (green) with the experimental structure (grey) (C$\alpha$ RMSD = 1.09 Å). (C) Comparison of the EC-based contact map of ZnT8 (black points) with contacts in the experimental (grey) and predicted (red) inward-open models. (D) Comparison of the EC-based contact map of ZnT8 (black) with contacts in the experimental (grey) and predicted (green) outward-open models. In panels C and D, major differences in the contact patterns of inward-open and outward-open states, supported by ECs unique to each state, are circled. Surface pockets for (E) inward-open and (F) outward-open states are represented as space-filled voids using the server https://kvfinder-web.cnpem.br/.

conformations of Znt8, as validated by comparison with experimental EC-derived contacts. A second test case for the ESM-AF2 modeling protocol using an SLC protein with both inward and outward-open experimental structures is presented for the *E. coli* D-galactonate:proton symporter (DgoT) in S1 Fig. Again, the ESM-AF2 protocol successfully modeled both inward- and outward-open states, consistent with experimental EC-derived contacts.

## Modeling alternative conformations of SLC proteins when a single experimental state is available

In the two cases above, we chose SLC proteins for which experimental structures of both outward- and inward-open conformations are available, and validated the ESM-AF2 modeling protocol against both the experimental atomic coordinates

(using Cα RMSD metrics) and against EC based contact maps, which are based on experimental primary sequence data. However, for most SLC proteins, experimental structures are only available for one (or neither) of the two states. We next modeled inward-open structures for two integral membrane proteins for which only the outward-open state is experimentally available. The results are shown in **Fig 3** for the 322-residue *Zea mays* CMP-sialic acid transporter 1 SLC35A1 (PDB id 6i1r-A [58]), a SLC35A subfamily member and in **Fig 4** for the 337-residue *Saccharomyces cerevisiae* GDP-mannose sugar transporter 1 Vrg4 (PDB id 5oge [59]), an SLC35D subfamily member. For both proteins, only outward-open X-ray crystal structures determined at 3.22 Å and 2.80 Å resolution, respectively, are available. In both of these cases, the ESM-AF2 protocol was not successful in providing models of the inward-open state that could be validated by patterns in the EC-based contact map unique to each conformer. However, using the ESM-MODELLER protocol, in which the outward-open state is modeled with AF2, and the inward-open state is modeled using a "flipped-sequence" as input to *ESMfold*, providing a virtual template that is then used with a conventional template-based modeling approaches, both outward- and inward-open states were generated. In both cases, the EC-based contact maps could be largely explained by the combined contact maps of these outward- and inward-open conformations, although some sporadic predicted ECs at the edge of the cutoff value used for identifying ECs were also present. These results validate the ESM-MODELLER

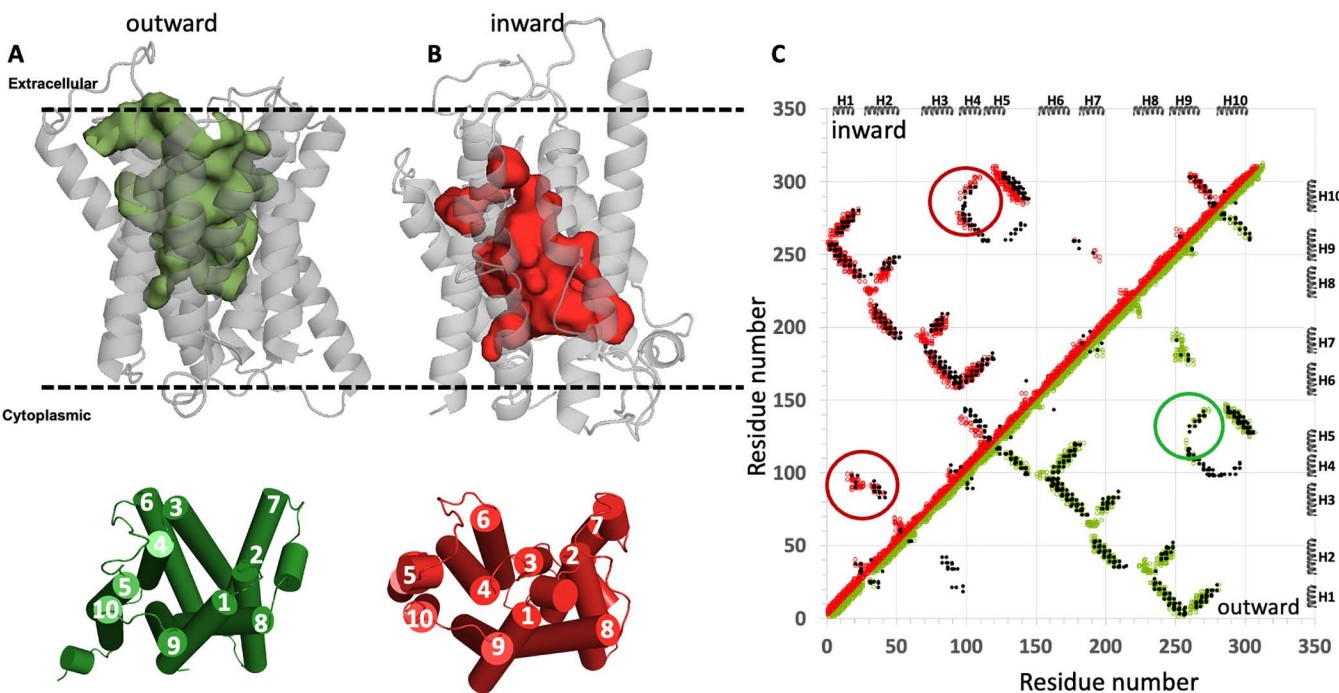

**Fig 3. ESM-MODELLER modeling of the inward-open conformation of the *Zea mays CMP-sialic acid transporter 1*.** (A) The experimental outward-open structure (PDB id 6i1r-A). (B) The inward-open structure modeled using ESM-MODELLER. In each of panels A and B the top images are ribbon representations of the protein structure with surface exposed cavities shown in either green (outward-open) or red (inward-open), and the bottom images are cylinder representations of these structural states with helices numbered 1 - 10. The dashed horizontal lines in panels A and B denote the approximate locations of the membrane boundaries. (C) The combined contact maps of the two resulting models are consistent with the experimental EC-based contact map. Green contacts are those present in the experimental outward-open model, and red contacts are those present in the predicted inward-open model. EC-based contacts are shown as black dots. The EC-based contacts circled in green are unique to the outward-open conformation, and those circled in red are unique to the inward-open conformation. At the thresholds chosen for ECs several predicted contacts are not explained by the combination of the two conformational states. In panels A and B (top), surface pockets are represented as space-filled voids using the server https://kvfinder-web.cnpem.br/.

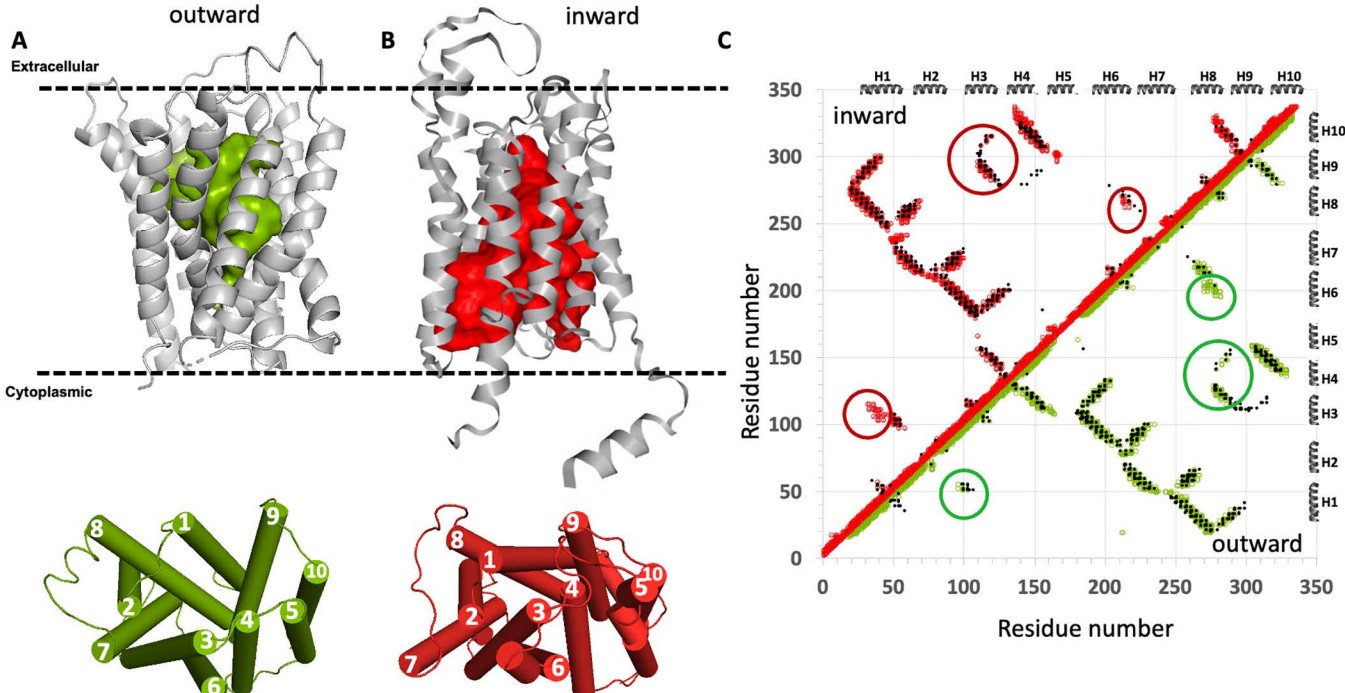

**Fig 4. ESM-MODELLER modeling of the inward-open conformation of the *S. cerevisiae* GDP-mannose sugar transporter 1, Vrg4.** (A) The experimental outward-open structure (PDB id 5oge). (B) The inward-open structure modeled using ESM-AF2. In each of panels A and B the top images are ribbon representations of the protein structure with surface exposed cavities shown in either green (outward-open) or red (inward-open), and the bottom images are cylinder representations of these structural states with helices numbered 1 - 10. The dashed horizontal lines in panels A and B denote the approximate locations of the membrane boundaries. (C) The combined contact maps of the two resulting models are consistent with the EC-based contact map. EC-based contacts are shown as black dots, inward-open contacts as red circles and outward-open contacts as green circles. The EC-based contacts circled in green are unique to the outward-open conformation, and those circled in red are unique to the inward-open conformation. At the thresholds chosen for ECs several predicted contacts are not explained by the combination of two conformational states. In panels A and B, surface pockets are represented as space-filled voids using the server https://kvfinder-web.cnpem.br/.

process for cases where, due to the impact of memorization of conformational states available at the time of training on the AF inference, the ESM-AF2 method fails.

## Modeling alternative conformations of SLC35F2 with ESM-AF2

Of particular interest are SLC proteins for which no experimental structures are available for either the inward- or outward-open states. SLC35F2 has < 12% sequence identity with any SLC35 subfamily members of known structure; in particular there is no good experimental structure that can be used as a template for comparative modeling of its inward- or outward-open conformations. Conventional AF2 modeling was carried out using the AF2-multimer colab server [55] executed both with the standard protocol without structural templates described in the Methods section and also with various other protocols using templates of distant homologues and multiple seeds. Modeling was also attempted using AF3 [51] without templates. Only the outward-open conformational state of SLC35 was returned by AF3. Hence, even without a state-specific structure in their training sets, AF2 (and AF3) are biased towards the outward-open state of SLC35F2.

For SLC35F2 we also explored using various protocols with shallow MSAs, dropouts, and the combination of dropouts with MSA masking to generate alternative conformational states. *AF-alt* was used to generate 480 models, and *AF_Sample* and *AF_Sample2* were used to generate 3,000 models each. These enhanced sampling methods are very GPU

intensive and require long run times. For this particular protein, for which no experimental structures were available in the PDB at the time of AF2 training, all three of these methods generated exclusively outward-open states (S2A–C Fig). These results for SLC35F2 illustrate the common case where even enhanced sampling methods fail to generate reliable models of multiple alternative conformational states. Interestingly, when *AF-Sample* was run on virtual flipped sequence of SLC35F2, exclusively inward-open conformational states for the flipped sequence were generated.

Having established the reliability, consistency, and limitations of the ESM-AF2 protocol, AF2 was used to model the outward-open conformation of SLC35F2, and both ESM-AF2 and ESM-Modeller were used to model its inward-open conformation (Fig 5). For EMS-AF2, the top-ranked model was outward-open, but other top-scoring models were inward-open. The contact maps of the resulting inward-open models generated by the two methods were then compared with their EC-based contact maps. The ESM-AF2 inward-open structure explains a few more EC-based contacts than the ESM-Modeller protocol, particularly for predicted contacts between helices H6 and H9 (*cf.* contact maps of Fig 5D and 5E). The excellent agreement between the EC-based contact map and combined contact maps of the computed outward- and inward-open structures validate the accuracy of the ESM-AF2 protocol for modeling this conformational variability of SLC35F2.

## Modeling alternative conformational states of other SLC proteins

We selected 4 additional SLC proteins for modeling with the ESM-AF2 and ESM-MODELLER protocol. These results are summarized in S3–S6 Figs. In all of these cases for which the structure of one conformational state was available in the PDB at the time of AF2 training, bias toward this state was observed when using AF2 alone or even when using AF2 with a template for the alternative state generated with ESM using a flipped sequence; i.e., the ESM-AF2 protocols described here fail to generate the alternative conformational state when one conformational state was available in the PDB at the time of AF2 training. However, the ESM-MODELLER protocol, which avoids the bias of conformational state modeling due to "memorization" often observed using AF2, provided models of both inward-open and outward-open states, with excellent agreement (< 1 - 2 Å rmsd) to experimental models where available, and in concordance with EC-predicted contact maps.

In carrying out the studies described above, we also assessed an array of protocols using the *ESMfold* models generated from a virtual flipped sequence as a template for modeling of the alternative conformational state followed by either AF2 or conventional template-based modeling. In this process, a shallow MSA was used so that the template structural information dominates the modeling process. AF2 modeling was done using single-sequence inference, and also with shallow MSAs (8, 16, or 32), recycle of 12, and with dropout. All 5 top-scoring models were assessed for representatives of the alternative conformational state. The original (e.g., outward-open) and final (e.g., inward-open) structures were validated by comparison against the EC-based contact map that will generally include predicted contacts for both conformational states. While the ESM-AF2 protocol could sometimes model the alternative conformational state, it was not successful in all cases (S3 Table). In all of these cases, the template-based modeling step of Fig 1, with the virtual flipped ESMfold structure as a template, could be performed successfully using MODELLER [52,56]. Template-based modeling could also be done using SwissModel [60] or other template-based modeling methods.

We also tested if the bias of AF2 to flip the conformational state of the ESM-generated template structure was also observed for AF3. This study is summarized in S4 Table. While the server version of AF3 does not support user-provided templates, the study could be done using a locally-installed version of AF3 with full MSAs generated with jackHMMR [61]. The same results were obtained using MSAs generated by MMseqs [62]. Considering the 6 proteins for which the ESM-AF2 protocol flipped the states of the input template, for two of these, reduced folate transporter SLC19A1 and thiamine transporter 1 SLC19A2, the AF3 protocol did not flip the state from the template; the overall ESM-AF3 protocol thus provided both states. However, for four others, aromatic amino acid exporter YddG, chloroquine resistance transporter I, *Zea mays* CMP-sialic acid transporter 1 SLC35A1, and *S. cerevisiae* GDP-mannose sugar transporter 1 SLC35D

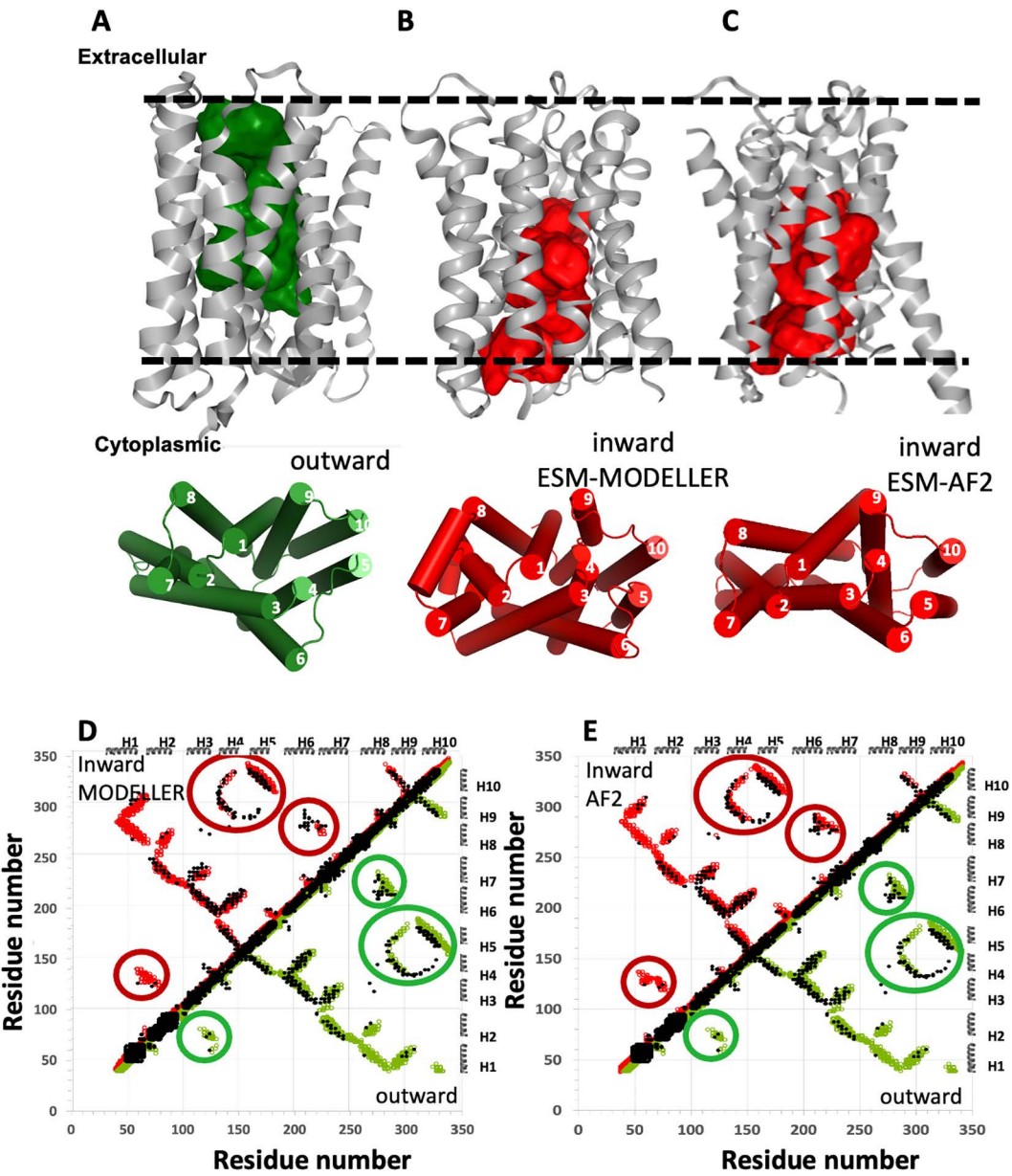

**Fig 5. Modeling of the outward- and inward-open conformations of human SLC35F2.** (A) The outward-open structure modeled with AF2. (B) The inward-open structure modeled using ESM-MODELLER. (C) The inward-open structure modeled using ESM-AF2. In each of panels A-C the top images are ribbon representations of the protein structure with surface exposed cavities shown in either green (outward-open) or red (inward-open), and the bottom images are cylinder representations of these structural states with helices numbered 1 - 10. The dashed horizontal lines in panels A and B denote the approximate locations of the membrane boundaries. (D) Contact maps of inward-open (red circles) and outward-open (green circles) models of panels A and B, and (E) contact maps of inward-open (red circles) and outward-open (green circles) models of panel A and C, superimposed on the EC contact map (black dots). In panels A, B and C, surface pockets are represented as space-filled voids using the server https://kvfinder-web.cnpem.br/.

Vrg4, the AF3 protocol, like the AF2 protocol, did flip the state from the template; and the overall ESM-AF3 protocol did not provide the alternative state. For the remaining two systems tested, both the ESM-AF2 and ASM-AF3 protocols were successfully guided by the ESM-generated template to successfully model the alterative state. Similar results were

obtained using shallow MSAs. On the other hand, for SLC35F2 using AF3 [51] with an inward-facing template generated by *ESMfold* returned only outward-facing models. Hence, the impact of training memorization on both the ESM-AF2 and ESM-AF3 protocols is highly variable and depends on multiple factors including the inference engine used and the nature and depth of the input MSA. These results demonstrate the importance of cross-validating models against EC or (if available) sample-specific experimental data.

## Discussion

We were very surprised to observe significant weaknesses of various published protocols using AF2 or AF3 for modeling alternative conformations of pseudo-symmetric SLC transporters. However, where conventional AF2/3 modeling (or even AF2 modeling with enhanced sampling) provides only one (either inward- or outward-open) conformational state; the alternative state can be modeled by the template-based ESM-AF (or ESM-MODELLER) protocol. The ESM-AF protocol is inspired by a more traditional approach using comparative modeling of the pseudo-symmetric halves of SLC transporters [31,44–50]. This traditional approach requires an accurate sequence alignment between the two symmetric halves of SLC protein to generate a structural template for the alternative state, which can be quite difficult to generate. In the ESM-AF (or ESM-MODELLER) approach, we use *ESMfold* to generate from a virtual flipped sequence a virtual protein structure, which is then used as a structure modeling template. Memorization bias does not significantly impact this process. This allowed us to reliably model alternative conformational states of several SLC transporters that were difficult to model using the traditional approach. Importantly, the resulting multi-state models are validated by comparison with sequence-based evolutionary co-variance data (ECs) that encode information about contacts present in the various conformational states adopted by the protein.

The ESM-AF2 approach is simple to implement and runs fast using publicly-available servers. Currently, the Deep-Mind AF3 server does not support template-based modeling, but ESM-AF3 can be run using locally-installed instances of AF3. However, despite the successful examples demonstrated in this study, the ESM-AF protocol for modeling alternative conformational states of pseudo-symmetric SLC proteins has some limitations. In particular, where structures of only one of the alternative states was available in the PDB at the time of training, a significant bias towards this state was observed when AF2/3 were used either directly or as part of the ESM-AF modeling process. Although this bias is overcome using the ESM-MODELLER protocol, it is somewhat disappointing to have to sometimes resort to older template-based modeling methods in place of AI-based methods like AF2/3. Another shortcoming is that neither protocol can be applied directly to homodimeric pseudo-symmetric SLC proteins, such as YiiP or EmrE [27,63]. Coordinates of SLC proteins with large loops and other structural decorations require manual editing to eliminate these loops/ decorations prior to applying the protocol. In addition, the validation of alternative state conformations by contact predictions relies on the quality of these contact predictions, and may not work well for SLC sequence families for which only shallow MSAs are available.

While we have focused our analysis on the outward and inward conformational states of SLC transporters, intermediate "occluded" states have also been captured in X-ray crystal and cryoEM structures. Although the ESM-AF2/MODELLER protocols sometimes also generate such occluded states, these states were not explored in this study. Ultimately, it would be desirable to develop methods which can generate complete conformational landscapes beyond the three major "inward-open", "occluded", "outward-open" states [12,64].

Symmetry in some integral membrane proteins such as SLC35F2 is easy to identify. Protein secondary structure and AF2 predictions indicate 10 helices connected by short loops. Chopping the sequence at the exact center of the loop between symmetric halves and swapping the coordinates of the two halves orients the helices with inverted conformation that overlays with the EC contact maps. The flipped sequence when used as input into *ESMfold* provides a flipped conformer which can be used as a template for further predictions as described in the Methods section. For SLC proteins that have a long loop between their symmetric halves, chopping the sequence in the exact center fails to provide a flipped sequence that can be used to successfully generate a proper flipped conformer. In these cases, the long loop was

replaced by an 8 to 12 residue polyglycine linker, and the split was made at the center of this linker. The resulting flipped sequence was sufficient for *ESMfold* to predict the flipped conformer. The contact maps generated for the flipped conformers are always compared with the EC-contact map. In cases where the match is not reflecting the contacts of the flipped state accurately, a few trials were made in the elimination of residues of the loop regions. It may also be possible to identify correct symmetry using repeat definitions identified by symmetry analyses available in the Encompass database [65].

In this study, alternative conformational states of SLC proteins were validated against contact predictions based on evolutionary co-variance (ECs). ECs are an interpretation of experimental protein sequence data, and hence this validation is effectively against experimental data. However, this validation is not as robust as validation against protein-specific experimental data such as X-ray crystallography, NMR, or cryoEM data.

The ability of AF2 to model protein structures not included in its training has been demonstrated in various CASP blind assessments [66,67]. Conventional AF2 was also reported to be successful in accurate modeling of protein structures determined by NMR methods which were not included in its training data, and for which no structures of homologous proteins were available at the time of training [68]. However, for proteins adopting multiple conformational states, AI training carried out with data that includes only one state can bias the inference and limit its ability to model the alternative state. Memorization can also result from homologous protein structures connected with the target protein through the MSA [9].

Recently Porter and co-workers have demonstrated that, at least for fold-flipping proteins which have significant structural differences between conformational states, AF2 modeling with enhanced sampling is often biased toward the conformational state reported in the PDB and potentially used in the AF2 training, and often is not able to predict conformational states not represented in the AF2 training data. Combining >280,000 models from several implementations of AF2 and AF3, only a 35% success rate was achieved in modeling alternative states of fold switchers for which one state was available for AF training [9,69]. In a related study of cryptic sites in proteins, Lazou et al. were able to use AF2 to generate both open and closed conformations for only 6 of 16 proteins studied [13], attributing this low success rate to bias due to training memorization. Bryant and Noé also have explored this question by training a structure prediction network, *Cfold*, on a conformational split of the PDB that excludes alternative conformations for protein structure pairs solved in two conformational states. While > 50% of experimentally-known nonredundant alternative protein conformations evaluated were predicted with high accuracy (TM-score > 0.8), for the remaining pairs *Cfold* failed to correctly model the alternative conformational state that was not included in the training data [11] and was biased toward the conformational state used in training. These results indicate that while in some cases, the network has learned enough to model alternative conformational states not included in the training data [70], in other cases success may in fact rely on some kind of memorization; i.e., both factors can be at play. *ESMfold* is generally less sensitive to this memorization bias [10]. Consistent with these observations, we also observed a bias toward previously reported conformational states when modeling with either AF2/3 or with the ESM-AF protocol outlined here. This bias was not suppressed by using single-sequences or very shallow MSAs (8–16 sequences) in the AF2/3 modeling. Nor was it overcome in the cases tested using enhanced sampling with AFSample [21] or AFSample2 [12]. For these SLC proteins, this bias is overcome using the ESM-MODELLER protocol. However, where no memorization bias is involved, the ESM-AF protocol is preferable as template-guided AF2 has more accurate properties than conventional template-based modeling methods.

While previous studies have demonstrated that training memorization bias impacts AF2 modeling of alternative states of "fold flip" proteins that have significant structural differences between the states [9], the *Cfold* study cited above, using a structure prediction network trained on a split version of the PDB excluding alternative conformational states of multi-state proteins [11] demonstrated less impact of memorization and high success rates in modeling distinct conformational states with smaller structural differences. The SLC proteins studied here have very similar overall structures and contact maps for the two states, yet the bias from conformational states available for the training process still strongly impacts the reliability of alternative conformational modeling by AF2/3. These observations suggest that it may be possible to use a

retrained AF network that excludes homologous structures from training data to suppress memorization bias and improve inference for modeling alternative conformational states of proteins.

## Conclusions

In this work we document bias in modeling multiple conformational states of SLC proteins that challenges the view that modeling of the multiple conformational states of this important class of integral membrane proteins is a largely solved problem. We describe, validate, and compare hybrid ESM-AF2, ESM-AF3, and ESM-MODELLER protocols for modeling alternative conformational states of pseudo-symmetric SLC proteins. This approach overcomes one shortcoming of conventional AF2/3 structure calculations which generally provide only one of the multiple conformational states observed experimentally. We observed that while AF2/3 generally does an excellent job of modeling one of the conformational states, there is a significant bias of AF towards conformational states available in the PDB at the time of its training. This bias can be overcome using the ESM-AF or ESM-MODELLER protocols. In this approach, the resulting multi-state models are validated by comparison with sequence-based EC data that encode information about contacts present in the various conformational states adopted by the protein. The method is simple to use, rapid to run, and can be implemented using public domain servers. Overall, the current study validates the ESM-AF/MODELLER protocol for modeling conformational heterogeneity of pseudo-symmetric SLC transporters, one of the most extensive class of transporters in the human proteome.

## Supporting information

**S1 Table. Assessment of modeling alternative states of SLC proteins with AF2.**
(DOCX)

**S2 Table. Assessment of successful generation of alternative states of SLC proteins from flipped sequence using *ESMfold* vs *AlphaFold2*.**
(DOCX)

**S3 Table. Modeling of both outward-open and inward-open states of pseudo-symmetric SLC proteins.**
(DOCX)

**S4 Table. Comparing bias of ESF-AF2, ESM-AF3, and ESM-MODELLER protocols in flipping the conformational state of SLC proteins.**
(DOCX)

**S1 Fig. Validation of ESM-AF2 protocol using an SLC protein with both outward- and inward-open experimental structures.**
(DOCX)

**S2 Fig. Conformational states of SLC35F2 modeled using various enhanced or massive sampling protocols.**
(DOCX)

**S3 Fig. Thiamine transporter 1 (SLC19A2).**
(DOCX)

**S4 Fig. Aromatic amino acid exporter YddG.**
(DOCX)

**S5 Fig. Reduced folate transporter (SLC19A1).**
(DOCX)

**S6 Fig. Chloroquine resistance transporter I.**
(DOCX)

**S7 Fig. Color-bind adjusted version of** Fig 2.
(DOCX)

**S8 Fig. Color-bind adjusted version of** Fig 3.
(DOCX)

**S9 Fig. Color-bind adjusted version of** Fig 4.
(DOCX)

**S10 Fig. Color-bind adjusted version of** Fig 5.
(DOCX)

**S11 Fig. Color-bind adjusted version of** S3 Fig.
(DOCX)

**S12 Fig. Color-bind adjusted version of** S4 Fig.
(DOCX)

**S13 Fig. Color-bind adjusted version of** S5 Fig.
(DOCX)

**S14 Fig. Color-bind adjusted version of** S6 Fig.
(DOCX)

## Acknowledgments

We thank Dr. Alberto Perez and Jokent Gaza for running inference using several structural templates with their locally-installed instance of *AlphaFold3*, and Dr. Davide Sala for providing scripts for running *AF-alt*. We also thank T.B. Acton, T. Benavides, A. De Falco, A. Gaur, R. Greene-Cramer, Y.J. Huang, T.A. Ramelot, B. Shurina, L. Spaman, and R. Tejero for helpful discussions and comments on the manuscript, and S. Collen for computer system administration support.

## Author contributions

**Conceptualization:** G.V.T. Swapna, Namita Dube, Monica J. Roth, Gaetano Montelione.

**Data curation:** G.V.T. Swapna.

**Formal analysis:** G.V.T. Swapna, Gaetano Montelione.

**Funding acquisition:** Monica J. Roth, Gaetano Montelione.

**Investigation:** G.V.T. Swapna, Namita Dube, Monica J. Roth, Gaetano Montelione.

**Methodology:** G.V.T. Swapna.

**Project administration:** Gaetano Montelione.

**Resources:** Gaetano Montelione.

**Supervision:** Monica J. Roth, Gaetano Montelione.

**Validation:** G.V.T. Swapna.

**Visualization:** G.V.T. Swapna, Gaetano Montelione.

**Writing – original draft:** G.V.T. Swapna, Namita Dube, Monica J. Roth, Gaetano Montelione.

**Writing – review & editing:** G.V.T. Swapna, Namita Dube, Monica J. Roth, Gaetano Montelione.

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
