## [Decision Letter · Decision Letter 0]

6 Jul 2025

PCOMPBIOL-D-25-00867

Memorization Bias Impacts Modeling of Alternative Conformational States of Solute Carrier Membrane Proteins with Methods from Deep Learning

PLOS Computational Biology

Dear Dr. Montelione,

Thank you for submitting your manuscript to PLOS Computational Biology. After careful consideration, we feel that it has merit but does not fully meet PLOS Computational Biology's publication criteria as it currently stands. Therefore, we invite you to submit a revised version of the manuscript that addresses the points raised during the review process.

Please submit your revised manuscript within 30 days Sep 05 2025 11:59PM. If you will need more time than this to complete your revisions, please reply to this message or contact the journal office at ploscompbiol@plos.org. Please include the following items when submitting your revised manuscript:

We look forward to receiving your revised manuscript.

Kind regards,

Alex Peralvarez-Marin

Academic Editor

PLOS Computational Biology

Arne Elofsson

Section Editor

PLOS Computational Biology

**Journal Requirements:**

At this stage, the following Authors/Authors require contributions: Gaetano Montelione. Please ensure that the full contributions of each author are acknowledged in the "Add/Edit/Remove Authors" section of our submission form.

5) We notice that your supplementary Figures, and Tables are included in the manuscript file. Please remove them and upload them with the file type 'Supporting Information'. Please ensure that each Supporting Information file has a legend listed in the manuscript after the references list.

2) If any authors received a salary from any of your funders, please state which authors and which funders.

7) Please provide a completed 'Competing Interests' statement, including any COIs declared by your co-authors. If you have no competing interests to declare, please state "The authors have declared that no competing interests exist". Otherwise please declare all competing interests beginning with the statement "I have read the journal's policy and the authors of this manuscript have the following competing interests:"

**Reviewers' comments:**

Reviewer's Responses to Questions

Reviewer #1: Swapna et al. used a template-based ESM modelling approach to obtain the alternate conformational states of SLC proteins. The authors also compared several ML based approaches such as AF2, AF3, ESM; highlighting the current limitations of these ML methods in modeling multiple conformational states of SLC proteins. The obtained structures were validated by using the EC data. The authors claimed to validate these structures using EC data as “experimental validation”, which I find surprising. Perhaps “validation” is more appropriate than “experimental validation”. The authors also highlighted the limitation of their high performing ESM-AF2 approach in modelling SLC proteins alternate states in several scenarios. The manuscript is well written and will be useful for researchers working on transmembrane proteins with multiple conformational states.

Minor points:

Page 13, line 3: “is it” should be “it is”

Page 13, 3rd paragraph: “Lazou al” should be “Lazou et al.”

Page 25, Fig. S3: The green and red circles seem to be flipped.

Reviewer #2: Though AlphaFold has revolutionized protein structure prediction, it has limitations. Here, Swapna and colleagues show one of its limitations–and some workarounds–when modeling alternative conformations of SLC Membrane proteins. The observations and approaches presented are useful, and I applaud the authors’ use of ECs to check the plausibility of the alternative conformations they generate. This work is solid and suitable for publication after a few minor concerns are addressed:

1) On Page 10, it is claimed that AF3 does not support template-directed modeling. What does this mean? AF3 uses templates in default mode, and it seems the input files could be modified to include templates of interest.

2) On Page 5, it says that AlphaFold-alt was run generating 30 conformations for sequence depths ranging from 16-32. Was it also run to generate MSAs with 4-8 sequences specified in Table S1? The Methods could be more clear on this point.

3) Faezov and Dunbrack also used templates to bias AlphaFold2 to generate models of alternative conformations of protein kinases (https://www.biorxiv.org/content/10.1101/2023.07.21.550125v2). Although they used a somewhat different approach than presented here, their work should be cited.

4) The manuscript seems to suggest that memorization occurs only when structures with the exact sequence being tested were in the database prior to training. This may be too narrow of a definition. AlphaFold can associate memorized structures with homologous sequences without evolutionary couplings (Chakravarty, et al. 2024). This is likely why, when CFold was trained (Bryant and Noé, 2024), all homologous alternative conformations were excluded from the training data. If one alternative structure is present in the training data, AF may associate its memorized structure with a homologous sequence without coevolutionary inference.

Reviewer #3: This is a very interesting article providing not only quantitative insights into the limitations of machine-learning methods in structure prediction of dynamic proteins, but also providing a robust, consistent strategy to predict alternate states for membrane proteins with inverted topologies. These proteins have repeats within their sequences, that the authors reverse and feed into ESMfold to create a template with alternate conformation that can then be used for template-based modeling of the alternate state. This is a creative extension of the method that I and others have developed in the past, while avoiding the pitfalls of repeat-based sequence alignments and also leveraging the high-quality models that are created by machine learning methods. The paper is generally clearly written and should be of value for the structure prediction community, especially since the scripts are publicly available.

I only have a few comments for the author's consideration:

- I wonder if "virtual flipped sequence" could/should be rephrased for clarity, perhaps "artificial" or "constructed" or "reordered" would be better terms. Essentially the repeated elements are swapped to create the new sequence.

- I could not find mention of how the sequence repeats are identified so that they can be swapped: Where one structure is known, the repeat definitions identified by symmetry analysis could be used, e.g. from the Encompass database.

- An desirable extension of the method would (eventually) be to create complete conformational landscapes beyond the three major "inward-open", "occluded", "outward-open" states. This should be mentioned in the discussion. Based on the ideas of del Alamo et al, we have recently been successful in applying AF2.3 with sequence-downsampling to create an extensive ensemble for a heterodimeric mitochondrial pyruvate carrier, as an example (PMID: 40249800). Could perhaps providing two templates as simultaneous inputs allow ESMfold to explore all those states?

- Several figures rely on red and green for comparisons; colorblind-friendly alternatives should be created

**Have the authors made all data and (if applicable) computational code underlying the findings in their manuscript fully available?**

Reviewer #1: Yes

Reviewer #2: Yes

Reviewer #3: Yes

PLOS authors have the option to publish the peer review history of their article (what does this mean? ). If published, this will include your full peer review and any attached files.

**Do you want your identity to be public for this peer review?** For information about this choice, including consent withdrawal, please see our Privacy Policy .

Reviewer #1: No

Reviewer #2: No

Reviewer #3: **Yes: ** Lucy Rachel Forrest

**Figure resubmission:**
---

## [Decision Letter · Decision Letter 1]

6 Oct 2025

Dear Prof Montelione,

We are pleased to inform you that your manuscript 'Memorization Bias Impacts Modeling of Alternative Conformational States of Solute Carrier Membrane Proteins with Methods from Deep Learning' has been provisionally accepted for publication in PLOS Computational Biology.

Best regards,

Alex Peralvarez-Marin

Academic Editor

PLOS Computational Biology

Arne Elofsson

Section Editor

PLOS Computational Biology

Reviewer's Responses to Questions

**Comments to the Authors:**

Reviewer #1: The authors addressed all the points.

Reviewer #3: The authors have addressed all my comments. I tank them for thoroughly considering the issue of color blind readers.

**Have the authors made all data and (if applicable) computational code underlying the findings in their manuscript fully available?**

Reviewer #1: None

Reviewer #3: Yes

PLOS authors have the option to publish the peer review history of their article (what does this mean? ). If published, this will include your full peer review and any attached files.

**Do you want your identity to be public for this peer review?** For information about this choice, including consent withdrawal, please see our Privacy Policy .

Reviewer #1: No

Reviewer #3: **Yes: ** Lucy R. Forrest

---

## [Editor Report · Acceptance letter]

PCOMPBIOL-D-25-00867R1

Memorization Bias Impacts Modeling of Alternative Conformational States of Solute Carrier Membrane Proteins with Methods from Deep Learning

Dear Dr Montelione,

I am pleased to inform you that your manuscript has been formally accepted for publication in PLOS Computational Biology. Your manuscript is now with our production department and you will be notified of the publication date in due course.

With kind regards,

Aiswarya Satheesan
